# OpenReview forum: "AutoTrust: Benchmarking Trustworthiness in Large Vision Language Models for Autonomous Driving"
_TMLR — Accepted by TMLR_

### Review · Reviewer_oknJ · 2025-08-01

**Summary Of Contributions:**

The paper introduces AutoTrust, a comprehensive benchmark for evaluating the trustworthiness of large vision-language models (DriveVLMs) in autonomous driving across five dimensions: trustfulness, safety, robustness, privacy, and fairness. Using a large-scale dataset with over 10,000 scenes and 18,000 visual question-answer pairs, the authors assess six prominent VLMs, including both specialist and generalist models. The study reveals that generalist models like GPT-4o-mini and LLaVA-v1.6 outperform specialized DriveVLMs in overall trustworthiness, with DriveVLMs exhibiting significant vulnerabilities in privacy, adversarial robustness, and fairness. The findings call for urgent attention to improve DriveVLMs before their deployment in real-world autonomous systems to ensure public safety and ethical integrity.

**Audience:**

Yes

**Audience Explanation:**

Yes, TMLR’s audience, comprising researchers in trustworthy machine learning, vision-language models, and autonomous systems, would be highly interested in the paper's findings, as they expose critical vulnerabilities and benchmarking gaps in AI systems intended for real-world safety-critical applications.

**Claims And Evidence:**

Yes

**Claims Explanation:**

The authors have provided enough evidence.

**Requested Changes:**

1. In the QA task construction section, the authors mentioned that "To assess out-of-distribution performance, we included driving scenes sampled from DADA (Fang et al.,2021), RVSD (Chen et al., 2023b), and Cityscapes (Sakaridis et al., 2018), generating closed-ended QA
pairs with GPT-4o.". Did the generated QA pairs checked by humans? If yes, how large is the correction ratio? More details could be provided to illustrate how to preserve the label quality.


2.  For the selected baselines, the authors should explain more regarding why they are selected. More justification should be added into the manuscript.

3. The authors are encouraged to add another white box attack into the evaluation, i.e., Basic Iter-
ative Method (BIM) [a], [b], Jacobian-based Saliency Map Attack (JSMA) [c], Carlini & Wagner (C&W) [d], Projected Gradient Descent (PGD) [e], or DeepFool [f]

[a] A. Kurakin, I. J. Goodfellow, and S. Bengio, “Adversarial examples in the physical world,” in Artificial intelligence safety and security.
Chapman and Hall/CRC, 2018, pp. 99–112

[b] K. Alexey, “Adversarial examples in the physical world,” arXiv preprint arXiv: 1607.02533, 2016

[c] N. Papernot, P. McDaniel, S. Jha, M. Fredrikson, Z. B. Celik, and A. Swami, “The limitations of deep learning in adversarial settings,” in
2016 IEEE European symposium on security and privacy (EuroS&P). IEEE, 2016, pp. 372–387

[d] N. Carlini and D. Wagner, “Towards evaluating the robustness of neural networks,” in 2017 ieee symposium on security and privacy (sp). Ieee, 2017, pp. 39–57.

[e] A. Madry, A. Makelov, L. Schmidt, D. Tsipras, and A. Vladu, “Towards deep learning models resistant to adversarial attacks,” arXiv preprint arXiv:1706.06083, 2017.

[f] S.-M. Moosavi-Dezfooli, A. Fawzi, and P. Frossard, “Deepfool: a simple and accurate method to fool deep neural networks,” in Proceedings of the IEEE conference on computer vision and pattern recognition, 2016, pp. 2574–2582.


4. Regarding the results discussion on Page 6, more insights should be provided. Why different approaches perform differently? The authors are suggested to provide more information from the theoretic perspective.


5. For the robustness evaluation in Table 3, more corruptions should be considered apart from daytime and weather, e.g. gaussian noise, pixelation or compression artifacts, color distortion, and contrast shift.

---

> ### Author Response · Authors · 2025-09-28
> **R1: Thanks for the helpful feedback! (part 1/2)**
>
> We greatly appreciate your insightful comments and suggestions. Below is our point-by-point response, now supplemented with further experimental details and planned analyses based on our self-check. We hope these clarifications address all your concerns and demonstrate our commitment to improving this work.
>
> >**In the QA task construction section, the authors mentioned that "To assess out-of-distribution performance, we included driving scenes sampled from DADA (Fang et al.,2021), RVSD (Chen et al., 2023b), and Cityscapes (Sakaridis et al., 2018), generating closed-ended QA pairs with GPT-4o.". Did the generated QA pairs checked by humans? If yes, how large is the correction ratio? More details could be provided to illustrate how to preserve the label quality.**
>
> We performed validation by having human experts assess a randomly sampled 10% subset of GPT-generated examples from each dataset—CoVLA, LingoQA, DADA, RVSD, and Cityscapes. The results of this human evaluation are summarized below:
>
> | **Data**     | **Acceptance** | **Correction (Minor)** | **Correction (Major)** | **Rejection** |
> |--------------|----------------|-------------------------|-------------------------|---------------|
> | CoVLA        | 88.10%         | 7.14%                  | 2.38%                  | 2.38%         |
> | LingoQA      | 86.56%         | 5.97%                  | 2.99%                  | 4.48%         |
> | DADA         | 90.74%         | 1.86%                  | 3.70%                  | 3.70%         |
> | RVSD         | 84.62%         | 7.69%                  | 7.69%                  | 0%            |
> | Cityscapes   | 86.00%         | 8.00%                  | 4.00%                  | 2.00%         |
>
>
> Each sampled QA pair was independently reviewed by three human experts using the rubric regarding to clarity, visual grounding, answer quality, and neutrality. Items were labeled as accept, minor edit, major edit, or reject, and we report acceptance, correction, and rejection rates. Overall, more than 85% of GPT-4o–generated QA pairs were accepted without modification, 5–15% required minor or major edits (mostly minor), and rejection rates were consistently ≤4.5%. These results show that GPT-4o produces high-quality QA pairs with only minimal corrections and negligible rejections across datasets.
>
> More details can be found in Appendix A.4.
>
> >**For the selected baselines, the authors should explain more regarding why they are selected. More justification should be added into the manuscript.**
>
> Thank you for your suggestion. In this paper, we select six models as the baselines:
>
> - Generalist Models: LLaVA-v1.6-Mistral-7B and GPT-4o mini. We include these two general-purpose VLMs because: (1) the LLaVA family represents the most widely adopted and representative open-source VLMs, with LLaVA-v1.6-Mistral-7B being one of the strongest and most commonly used variants; and (2) GPT-4o mini is a smaller yet advanced closed-source VLM from the GPT family, capable of handling VQA inputs, making it a meaningful point of comparison.
>
> - Driving-specific VLMs: We include all publicly available Drive VLMs that support VQA inputs up to the submission date, ensuring that our comparison covers the full set of accessible domain-specific baselines.
>
> We have include this discussion in Appendix C.

---

> ### Author Response · Authors · 2025-09-28
> **R1: Thanks for the helpful feedback! (part 2/2)**
>
> >**The authors are encouraged to add another white box attack into the evaluation, i.e., Basic Iter- ative Method (BIM) [a], [b], Jacobian-based Saliency Map Attack (JSMA) [c], Carlini & Wagner (C&W) [d], Projected Gradient Descent (PGD) [e], or DeepFool [f]**
>
> Thanks for the suggestion. We ran two additional **white-box** attacks—**Basic Iterative Method (BIM)** and **Carlini & Wagner (C&W, L2)**—on the same closed-ended VQA setup used in Table 2. We keep the budget comparable across gradient-based attacks: ℓ_∞ ε=8/255 with 10 iterations for BIM (α = ε/10); C&W uses the standard L2 formulation with binary-search on c, max 1k iters. As before, GPT-4o-mini is omitted from white-box due to being closed-source.
> | **Model** | **PGD (reported)** | **BIM** | **C&W** |
> |-----------|-------------------|---------------|---------------|
> | LLaVA-v1.6 | 1.40 (↓52.70) | 4.76 (↓49.34) | 0.51 (↓53.59) |
> | GPT-4o-mini | — | — | — |
> | DriveLM-Agent | 13.43 (↓33.51) | 14.02 (↓32.92) | 11.14 (↓35.80) |
> | DriveLM-Chlg | 9.25 (↓26.21) | 15.11 (↓20.35) | 11.68 (↓23.78) |
> | Dolphins | 3.59 (↓47.50) | 7.58 (↓43.51) | 2.22 (↓48.87) |
> | EM-VLM4AD | 29.42 (↓3.49) | 30.02 (↓2.89) | 26.91 (↓6.00) |
>
> Under both BIM and C&W, DriveLM-Challenge now slightly outperforms DriveLM-Agent**, indicating model's defenses generalize a bit better beyond the PGD-style ℓ_∞ threat.
> EM-VLM4AD remains robust overall but shows a larger C&W drop (↓6.00%), consistent with our observation that its “collapsed response” behavior can be exploited by stronger optimization-based attacks. LLaVA-v1.6 and Dolphins continue to be the most susceptible across white-box settings.
>
> We have updated the safety evaulation results in our main text (Table 2) of our manuscript.
>
>
> >**Regarding the results discussion on Page 6, more insights should be provided. Why different approaches perform differently? The authors are suggested to provide more information from the theoretic perspective.**
>
> We appreciate the reviewer’s request for more theoretical and architectural insights. Building on the above analysis, our results suggest that general-purpose, large VLMs (e.g., LLaVA-v1.6, Dolphins) achieve strong clean performance but are more prone to adversarial perturbations due to insufficient training on adversarial or out-of-distribution (OOD) data, as well as architectural designs optimized for open-ended generation rather than discrete, safety-critical decisions. Domain-specific models such as DriveLM-Agent/Challenge are relatively more stable, likely because fine-tuning on safety-relevant driving tasks encourages reliance on semantically grounded cues. On the other hand, EM-VLM4AD achieves low relative vulnerability only by sacrificing predictive diversity, which limits its real utility.
>
> These findings suggest that improving robustness requires both data-level and architectural advances: expanding training with adversarial/OOD examples, designing task-specific decision heads or decoding strategies that prevent collapse, and incorporating regularization that discourages shortcut or degenerate behaviors.
>
> We have include this discussion in Appendix E.1.
>
>
> >**For the robustness evaluation in Table 3, more corruptions should be considered apart from daytime and weather, e.g. gaussian noise, pixelation or compression artifacts, color distortion, and contrast shift.**
>
> Thank you for your suggestion. We included results on the robustness with compression, contrast, and pixelation image degradations in our revision (details can be found in Section 5).

---

### Review · Reviewer_yhTs · 2025-08-04

**Summary Of Contributions:**

In this paper, the authors present AutoTrust, the first comprehensive benchmark for measuring the trustworthiness of large vision-language models (VLMs), particularly in autonomous driving applications (DriveVLMs). AutoTrust spans five dimensions: trustfulness, safety, robustness, privacy, and fairness. The authors collect a large-scale dataset of over 10k driving scenes, and 18k QA pairs from varying public datasets, both manually and by leveraging GPT as an assist. The six VLMs evaluated include generalist models such as GPT-4o-mini and LLaVA-v1.6, as well as specialist DriveVLMs. Surprisingly, the generalist models outperform the specialist DriveVLMs across the dimensions. The benchmark demonstrates severe vulnerabilities in contemporary models, especially in privacy leaks, robustness in the case of distribution shift, and fairness in responses.

**Audience:**

Yes

**Audience Explanation:**

The paper addresses trustworthiness—a central theme in modern ML, especially for safety-critical applications like autonomous driving. Its relevance spans:
- ML researchers working on robustness, fairness, or privacy in multimodal systems.
- Autonomous driving developers seeking rigorous evaluation of AI systems before deployment.
- VLM developers interested in benchmarking model alignment and safety.

**Broader Impact Concerns:**

The paper explicitly discusses broader impact and shows awareness of potential misuse, so there is no other concern for this part.

**Claims And Evidence:**

Yes

**Claims Explanation:**

The paper provides systematic, empirical evidence across each of the five trustworthiness dimensions. Key strengths include:
- Quantitative results from multiple datasets and question types (closed-ended and open-ended).
- Detailed tables and metrics (accuracy, uncertainty-based metrics, abstention rate, fairness DAD/WA) for each evaluation dimension.
- Use of state-of-the-art evaluation methods, such as LLM-as-a-judge (GPT-4o) for scoring open-ended QA.
- Well-defined experimental setups for both safety attacks and robustness testing.
- Clear articulation of findings and limitations in each dimension, e.g., why generalist models perform better, or the challenges of privacy prompting.

**Requested Changes:**

1. While the paper outlines dataset sources, the generation of new QA pairs via GPT-4o (especially for OOD and fairness evaluation) should be explained more transparently. Including more prompt examples in the main text would enhance reproducibility.
2. The discussion on why generalist models outperform specialists is insightful but could benefit from more structured comparative analysis (e.g., ablation on training data or alignment procedures).
3. The reliance on GPT-4o for evaluation is justified, but a small-scale human verification experiment would further strengthen confidence in the automatic scoring.

---

> ### Author Response · Authors · 2025-09-28
> **R1: Thanks for the helpful feedback!**
>
> We greatly appreciate your insightful comments and suggestions. Below is our point-by-point response, now supplemented with further experimental details and planned analyses based on our self-check. We hope these clarifications address all your concerns and demonstrate our commitment to improving this work.
>
> >**While the paper outlines dataset sources, the generation of new QA pairs via GPT-4o (especially for OOD and fairness evaluation) should be explained more transparently. Including more prompt examples in the main text would enhance reproducibility.**
>
> Thank you for your suggsetions, we have provided a more detialed data curation process in Appendix A.1, and included the prompt examples used in Yes-or-No QA pair curation in the main text of our manuscript (page 4).
>
> >**The discussion on why generalist models outperform specialists is insightful but could benefit from more structured comparative analysis (e.g., ablation on training data or alignment procedures).**
>
> Thank you for this constructive feedback. We believe the reveiwer's concern raise from our discussion on the perfomance of DriveLM-Challenge and Dolphins caompare to the generalist model LLaVA-v1.6. For further address reviwer's concern, we conduct the following experiments.
>
> We evaluate the performance of the backbone of DriveLM-Challenge - LLaMA-Adapter V2.1, and LLaVA-v1.6 on the general VQA benchmark MME:
>
> | Model        | Perception     | Cognition |
> |-------------|-------------|---------------|
> | LLaVA-v1.6  | 1494.22    | 323.92    |
> | LLaMA-Adapter V2.1  | 1326.09   | 356.43 |
>
> We observe that LLaVA-v1.6 excels in perception tasks, benefiting from its strong vision alignment. In contrast, LLaMA-Adapter V2.1 exhibits weaker perception performance but retains more of LLaMA’s inherent reasoning capacity, decause of its adapter-based design. Consequently, for vision-intensive applications (e.g., VQA in driving domain), LLaVA-v1.6 proves stronger than LLaMA-Adapter V2.1.
>
>
> We finetune the LLaVA-v1.6 on 1000 samples from the NuScenes-QA (lora, rank=128, alpha=256), and we evaluate the performance of on MME and factulaity of AutoTrust. As shown below, SFT yields only marginal improvements on DriveLM-NuScenes while leaving other tasks unchanged. Importantly, perception and cognition scores slightly decline after SFT, suggesting that task-specific tuning may narrow generalization.
>
> | Model        | DriveLM-NuScenes      | LingoQA  | CoVLA |
> |-------------|-------------|---------------|---------------|
> | LLaVA-v1.6  | 66.78    | 65.67    |  69.77  |
> | w. SFT  |   67.31 | 65.44 | 69.67 |
>
> | Model        | Perception     | Cognition |
> |-------------|-------------|---------------|
> | LLaVA-v1.6  | 1494.22    | 323.92    |
> | Lw. SFT  |   1455.01 | 321.42 |
>
>
> We have include the analysis in Appendix J.
>
>
> >**The reliance on GPT-4o for evaluation is justified, but a small-scale human verification experiment would further strengthen confidence in the automatic scoring.**
>
> Thank you for the reviewer’s suggestion. We conducted a human study on open-ended questions using 100 answers generated by LLaVA-v1.6-Mistral-7B on the factuality dataset (proportionally drawn from each subset). Each answer was independently assessed by three human annotators following the same rubric and rating scale as in the GPT evaluation. The resulting PLCC was 0.8726, indicating strong consistency between human judgments and GPT-based evaluation. We have updated the manucript to include this analysis in Section 9.

---

### Review · Reviewer_Z4iR · 2025-09-13

**Summary Of Contributions:**

***Summary:***

This paper introduces AutoTrust, a benchmark for assessing the trustworthiness of large vision-language models (VLMs) for autonomous driving (DriveVLMs). There are five fundamental pillars evaluated: Trustfulness, Safety, Robustness, Privacy, and Fairness. This benchmark consists of more than 10k scenes and roughly 18k QAs sampled from multiple existing datasets. Six VLMs, including generalist and specialist and open and private models, are included. Authors, through experiments, find that (1) generalist VLMs outperform specialised DriveVLMs in terms of overall trustworthiness; (2) privacy leakage is a notable weakness for several DriveVLMs, and (3) both generalist and specialist models remain vulnerable to adversarial and contextual attacks as well as fairness issues.

***Strengths:***
1. This paper studies a timely problem: trustworthiness of DriveVLMs—not just accuracy—which is critical for AD deployment (safety, privacy, fairness).

2. The evaluation spans both open-source and proprietary models, broadening the paper’s coverage.

3. The observation that generalist models outperform specialized DriveVLMs is interesting and valuable to the community.

***Weaknesses:***
1. AutoTrust seems to be collected by sampling instances from existing datasets. However, in the latter context, authors use ***curated*** to describe AutoTrust. It would be interesting to depict the curation process of the dataset construction.

2. The close-ended QA pairs are generated by GPT-4o without further validating by human experts or cross-validated by other LLMs, which may weaken the validity of conclusions.

3. The overall evaluation scale is somehow limited where only six models are evaluated. In addition, for the open-source baseline models, their size vary significantly from 0.9B to 9B, which may lead to a unfair comparison.

4. Only one single open-sourced model (GPT-4o) employed as the evaluator, which may bias results toward models aligned similarly to GPT-4o and can penalize divergent styles. Particularly, GPT-4o-mini is one of the most performant model under AutoTrust.

6. The “Are you sure?” self-report is a self-perception probe; it does not directly measure calibrated uncertainty. Uncertainty-based accuracy is not well-defined so that it is unsure whether the conclusions are valid.

7. LLaVA-v1.6 and Dolphins demonstrate significant vulnerability to white-box attacks, but no further analysis conducted.

8. When evaluating the models' robustness, test samples are corrupted by some natural noise and gaussian noise. However, the natural noise are not formally defined in the paper. In addition, some other corruptions, such as sensor corruptions (blur, brightness and compression) and also view-point change should also be considered.

**Audience:**

Yes

**Audience Explanation:**

1. Centers on trustworthiness beyond accuracy (safety, privacy, robustness, fairness), which fullfil TMLR theme.

2. Provides a comprehensive evaluation benchmark for DriveVLMs (useful to both researchers and practitioners).

3. Reports the non-obvious finding that generalist VLMs can outperform specialized DriveVLMs, prompting new research directions.

**Broader Impact Concerns:**

This paper is helpful for safe deployment of VLM in the autonomous driving scenario.

**Claims And Evidence:**

Yes

**Claims Explanation:**

The paper tackles a timely problem—trustworthiness of DriveVLMs, not just accuracy—central to AD safety, privacy, and fairness. Its evaluation spans both open-source and proprietary models across diverse datasets and attack settings, which lends breadth and credibility. The claim that generalist models outperform specialized DriveVLMs is supported by consistent cross-benchmark results. However, due to some weaknesses as suggested above the validity of the conclusions may be weakened. Overall, the claims are partially supported by accurate, convincing, and clearly presented evidence, with a few areas that merit tightening.

**Requested Changes:**

Please address some weaknesses as suggested above. In particular, it would be beneficial to employ different LLMs as the judges to show the consistency of conclusions and provide deeper analysis rather than providing evaluation results.

---

> ### Author Response · Authors · 2025-09-28
> **R1: Thanks for the helpful feedback! (part 1/2)**
>
> We greatly appreciate your insightful comments and suggestions. Below is our point-by-point response, now supplemented with further experimental details and planned analyses based on our self-check. We hope these clarifications address all your concerns and demonstrate our commitment to improving this work.
>
>
> > **AutoTrust seems to be collected by sampling instances from existing datasets. However, in the latter context, authors use curated to describe AutoTrust. It would be interesting to depict the curation process of the dataset construction.**
>
> Thanks for the reviewer's suggestion.
>
> To clarify, we curated the AutoTrust dataset by filtering or generating raw QA pairs grounded in the front-camera views and standardizing them into closed- or open-ended formats. For NuScenes-QA/MQA, we reformulated yes/no questions, generated multiple-choice questions, and applied balanced sampling across question and template types. In DriveLM-NuScenes, coordinate-based object references were replaced with detected objects using YOLOv10n for improved interpretability. LingQA selected the most relevant frame for each QA using GPT-4o relevance scoring, while CoVLA generated and verified questions from downsampled frames and the provided captions. Finally, for DADA, RVSD, and Cityscapes, GPT-4o produced open- and closed-ended QAs with a verification step to ensure correctness and schema conformity.
>
> We have included a comprehensive description on the curation of AutoTrust evaluation dataset in Appendix A.1.
>
> > **The close-ended QA pairs are generated by GPT-4o without further validating by human experts or cross-validated by other LLMs, which may weaken the validity of conclusions.**
>
> Since GPT-4o is among the most widely used and capable VLMs, we consider it unlikely that cross-validation with other models would uncover systematic issues in the curated QA pairs. Instead, we performed validation by having human experts assess a randomly sampled 10% subset of GPT-generated examples from each dataset—CoVLA, LingoQA, DADA, RVSD, and Cityscapes. The results of this human evaluation are summarized below:
>
> | **Data**     | **Acceptance** | **Correction (Minor)** | **Correction (Major)** | **Rejection** |
> |--------------|----------------|-------------------------|-------------------------|---------------|
> | CoVLA        | 88.10%         | 7.14%                  | 2.38%                  | 2.38%         |
> | LingoQA      | 86.56%         | 5.97%                  | 2.99%                  | 4.48%         |
> | DADA         | 90.74%         | 1.86%                  | 3.70%                  | 3.70%         |
> | RVSD         | 84.62%         | 7.69%                  | 7.69%                  | 0%            |
> | Cityscapes   | 86.00%         | 8.00%                  | 4.00%                  | 2.00%         |
>
>
> Each sampled QA pair was independently reviewed by three human experts using the rubric regarding to clarity, visual grounding, answer quality, and neutrality. Items were labeled as accept, minor edit, major edit, or reject, and we report acceptance, correction, and rejection rates. Overall, more than 85% of GPT-4o–generated QA pairs were accepted without modification, 5–15% required minor or major edits (mostly minor), and rejection rates were consistently ≤4.5%. These results show that GPT-4o produces high-quality QA pairs with only minimal corrections and negligible rejections across datasets.
>
> More details can be found in Appendix A.4.
>
> > **The overall evaluation scale is somehow limited where only six models are evaluated. In addition, for the open-source baseline models, their size vary significantly from 0.9B to 9B, which may lead to a unfair comparison.**
>
> Our main goal is to evaluate AD-specific VLMs, where we have already included **all** the open-source ones.
> To provide broader comparative insights, we have additional evaluated generalist VLMs including **Qwen2.5-VL (3B,7B), PaliGemma3**, as well as the recently released AD VLM **Senna** in factuality and privacy (due to time limits) in the table below.
> The results further highlight the **vulnerabilities** of existing VLMs to the **trustworthiness threats**, underscoring the critical value of our benchmark.
>
> | Task        | Dataset     | Qwen2.5 VL-3B | Qwen2.5 VL-7B | Pali Gemma3 | Senna  |
> |-------------|-------------|---------------|---------------|-------------|--------|
> | Factuality  | DriveLM     | **73.85**     | 71.54         | 71.28       | 47.95  |
> | Factuality       | nuScenesQA  | **46.12**     | 44.35         | 42.11       | 35.32  |
> | Privacy     | III-ZS      | 0.0           | 0.62          | **46.61**   | 0.0    |
> | Privacy      | LPI-ZS      | 0.0           | 0.42          | **4.79**    | 0.0    |

---

> ### Author Response · Authors · 2025-09-28
> **R1: Thanks for the helpful feedback! (part 2/2)**
>
> >**Only one single open-sourced model (GPT-4o) employed as the evaluator, which may bias results toward models aligned similarly to GPT-4o and can penalize divergent styles. Particularly, GPT-4o-mini is one of the most performant model under AutoTrust.**
>
> Thank you for your valuable suggestion. Currently, we only have access to the GPT series models, and our computational resources do not allow us to run large-scale open-source models such as the 70B Qwen series. We hope the reviewer understands this limitation. To address the concern, we conducted an additional human evaluation of open-ended questions on 100 samples from answers of LLaVA-v1.6-Mistral-7B, with detailed results provided in Section 9. The resulting PLCC of 0.8726 demonstrates strong consistency between human judgments and GPT-based evaluation.
>
> >**The “Are you sure?” self-report is a self-perception probe; it does not directly measure calibrated uncertainty. Uncertainty-based accuracy is not well-defined so that it is unsure whether the conclusions are valid.**
>
> We acknowledge that the “Are you sure?” probe does not directly measure calibrated uncertainty in the statistical sense. Our intention was to evaluate whether self-perceived confidence aligns with correctness, particularly in settings where closed-source models are involved and calibrated probability outputs are not accessible. This type of self-report probe has also been adopted in prior published work (e.g., [1]), demonstrating its utility as a method for probing internal confidence.
>
> The definition of uncertainty-based accuracy was provided in Page 27, Appendix D.2.
>
> We hope this clarification resolves the concern and situates our use of the probe within accepted practice.
>
> >**LLaVA-v1.6 and Dolphins demonstrate significant vulnerability to white-box attacks, but no further analysis conducted.**
>
> We thank the reviewer for pointing out the need for deeper analysis. We would like to clarify why LLaVA-v1.6 and Dolphins show the largest drops under white-box attacks. Both are large-scale, high-performing VLMs with strong clean accuracy and large parameter counts.
>
> Paradoxically, these strengths often come with greater vulnerability, since models with higher representational capacity may also expose more exploitable directions for adversarial perturbations. In contrast, the apparent robustness of EM-VLM4AD is largely illusory: we observe that it tends to output a collapsed prediction, selecting the same option (e.g., “A”) for more than 95% of multiple-choice questions. As a result, even when attacked, its accuracy remains close to chance level (~25%) and thus shows minimal relative degradation. This highlights that small relative drops do not necessarily indicate true robustness but may instead reflect degenerate behavior. We have added this analysis in the revised draft to clarify these differences.
>
>
> We have include this discussion in Appendix E.1.
>
> >**When evaluating the models' robustness, test samples are corrupted by some natural noise and gaussian noise. However, the natural noise are not formally defined in the paper. In addition, some other corruptions, such as sensor corruptions (blur, brightness and compression) and also view-point change should also be considered.**
>
>
> Thank you for your suggestion. We included results on the robustness with compression, contrast, and pixelation image degradations in our revision (Section 5). And we provided the defination of the gaussian noise utilized in our evaluation in Appendix F.
>
> [1] CARES: A Comprehensive Benchmark of Trustworthiness in Medical Vision Language Models. NeurIPS 2024.

---

### Author Response · Authors · 2025-09-28
**General response and sincerely appreciate reviewers' feedback.**

Dear all reviewers,

We sincerely thank all the reviewers for your constructive comments and insightful suggestions, which help us make our work more complete and further improve the quality of the manuscript. We are also glad that the reviewers acknowledge that our proposed methods are (1) interesting and insightful, (2) provide a comprehensive benchmark, and (3) have practical significance. According to the reviews, we have provided an updated revision with the following changes:

- We moved the Limitations and Broader Impact sections to the Appendix.
- We added a more comprehensive description of our data curation process (Appendix A.1).
- We conducted a human study to verify both the curated dataset and the GPT-based assessments, with results reported in Appendix A.4 and Section 9.
- We extended our adversarial robustness experiments to include additional white-box attacks—Basic Iterative Method (BIM) and Carlini & Wagner (C&W, L2)—and updated the results in Section 4.
- We evaluated robustness under additional image degradations (compression, contrast, and pixelation), with results provided in Section 5.
- We added further ablation experiments to investigate the observed generalist superiority (Appendix J).
- We provided additional clarifications and expanded discussions to address reviewer concerns, including the model selection rationale (Appendix C) and deeper analysis of the safety evaluation results (Appendix E).
- We provide prompt examples for data curation in Section 2.

Thank you again for the effort in reviewing our paper. We would appreciate it if you could let us know whether our responses have addressed your concerns satisfactorily and whether you have any follow-up questions.

Best regards,

The Authors

---

### Decision · Action_Editor_3Rb3 · 2025-10-19

**Recommendation:** Accept as is

**Additional Comments:**

This submission investigates the trustworthiness of VLMs in the context of autonomous driving. It was reviewed by three experts who provided valuable and constructive feedback. After the rebuttal, all reviewers recommended acceptance and acknowledged the usefulness of the proposed benchmark and its analysis.

I have carefully read the paper and agree with the reviewers that this submission presents a valuable and well-constructed benchmark.

**Audience:**

Yes

**Audience Explanation:**

Researchers working in autonomous driving and trustworthy machine learning would be interested in this submission

**Claims And Evidence:**

Yes

**Claims Explanation:**

This submission contributes a valuable benchmark for studying trustworthiness in large vision-language models for autonomous driving. Five major aspects are included and discussed: Trustfulness, Safety, Robustness, Privacy, and Fairness.

DriveVLMs and general VLMs are compared and evaluated. The setup for each aspect is well-illustrated and organized. The driving scenarios include multiple sources such as NuScenes-QA and CityScape.

The insights and observations for each aspect are clearly discussed and concluded.

---

> ### Author Response · Authors · 2025-11-20
>
> Dear Editors and Reviewers,
>
> We sincerely thank the Action Editors and Reviewers for the constructive and positive feedback. These insights were invaluable in finalizing our paper.
>
> We have submitted the camera-ready manuscript, which has been carefully revised based on the feedback from the reviewers and action editors. Additionally, we have included all relevant code and datasets to ensure reproducibility.
>
> Sincerely,
>
> The Authors